# The Role of Adaptogens in Prophylaxis and Treatment of Viral Respiratory Infections

**DOI:** 10.3390/ph13090236

**Published:** 2020-09-08

**Authors:** Alexander Panossian, Thomas Brendler

**Affiliations:** 1Phytomed AB, Vaxtorp, 31275 Halland, Sweden; 2EuropharmaUSA, Green Bay, WI 54311, USA; 3Department of Botany and Plant Biotechnology, University of Johannesburg, Johannesburg 2000, South Africa; txb@plantaphile.eu; 4Traditional Medicinals Inc., Rohnert Park, CA 94928, USA; 5Plantaphile, Collingswood, NJ 08108, USA

**Keywords:** adaptogens, *Andrographis*, *Eleutherococcus*, *Glycyrrhiza*, *Panax*, *Rhodiola*, *Schisandra*, *Withania*, melatonin, viral infection

## Abstract

The aim of our review is to demonstrate the potential of herbal preparations, specifically adaptogens for prevention and treatment of respiratory infections, as well as convalescence, specifically through supporting a challenged immune system, increasing resistance to viral infection, inhibiting severe inflammatory progression, and driving effective recovery. The evidence from pre-clinical and clinical studies with *Andrographis paniculata*, *Eleutherococcus senticosus*, *Glycyrrhiza* spp., *Panax* spp., *Rhodiola rosea*, *Schisandra chinensis*, *Withania somnifera*, their combination products and melatonin suggests that adaptogens can be useful in prophylaxis and treatment of viral infections at all stages of progression of inflammation as well as in aiding recovery of the organism by (i) modulating innate and adaptive immunity, (ii) anti-inflammatory activity, (iii) detoxification and repair of oxidative stress-induced damage in compromised cells, (iv) direct antiviral effects of inhibiting viral docking or replication, and (v) improving quality of life during convalescence.

## 1. Introduction

The COVID-19 pandemic brought new challenges to biomedical sciences, specifically, the development of effective therapeutics for prevention and treatment of acute viral and stress-induced diseases. Unfortunately, the potential of herbal preparations in prevention and treatment of viral infections is underestimated. Lack of solid evidence for efficacy and safety from randomized, controlled clinical studies is often cited as a reason for dismissal. In reality, the risk of adverse events is significantly higher for synthetic antiviral and immunotropic drugs than for the vast majority of herbal preparations. The COVID-19 pandemic, for which to date no cure or vaccine exist, thus provides a more than timely context in terms of findings related to epidemiology and pathogenesis in which to discuss relevant evidence from pre-clinical and clinical studies of herbal preparations, specifically adaptogens.

Pathogenesis and progression of a viral infection is a multistep process [1,2], which requires an appropriate therapeutic strategy starting with initiation of overall defense response to the pathogen [3,4,5,6]. Included in this process are numerous extra- and intracellular interactions between components of host defense and life cycle regulation systems on all levels of regulation—genomic, transcriptomic, proteomic, metabolomic and macrobiotic [7]. Consequently, effective prevention or treatment of a viral infection and other viral infections requires pharmaceutical intervention affecting the innate and adaptive immune system, phases I–III metabolizing enzymes of detoxifying and repair systems, as well as the virus’ life cycle and proliferation (Figure 1). This can be achieved with herbal preparations that have polyvalent and pleiotropic actions on host defense systems. For instance, it was found that more than half of SARS-CoV-2-infected subjects were asymptomatic at the time of testing [8], which points at the ability of the innate immune system to curb progression of COVID-19 at an early stage of invasion of the pathogen. Both activation and inhibition of various components of innate immune system [4,5,9] by numerous natural compounds of plant kingdom is well documented in many publications. Specifically, complex mixtures of natural compounds (or herbal extracts) synergistically targeting multiple elements of molecular networks involved in inflammatory defense response are presumably more effective than mono-drugs that target only one receptor [10,11].

Adaptogens are natural stress-protective compounds or plant extracts that increase adaptability, resilience, and survival of organisms [12]. Adaptogens increase “the state of non-specific resistance” of organisms [13] to harm [14,15], including bacterial and viral pathogens. Non-specific defense responses to pathogens depend on the body′s ability to recognize conserved features of pathogens by the evolutionarily ancient innate immune system, a group of proteins and phagocytic cells, which become activated during the critical first hours and days of infection to destroy invaders [9]. The basic mechanisms of innate immune responses that regulate innate defense, e.g., pattern recognition by toll-like receptors (TLR), defensins, etc., are conserved and apparently involved in innate immunity in all multicellular organisms. Their conservation during evolution shows the importance of innate responses in the defense against microbial and viral pathogens [9].

More than 100 medicinal plants have been reported to have adaptogenic activity, however only few, i.e.,

*Andrographis paniculata* (Burm. F.) Wall. ex. Nees, Acanthaceae (AP),*Eleutherococcus senticosus* (Rupr. & Maxim.) Maxim, Araliaceae (ES),*Glycyrrhiza* spp., Fabaceae (GS),*Panax* spp., Araliaceae (PS),*Rhodiola rosea* L., Crassulaceae (RR),*Schisandra chinensis* (Turcz.) Bail., Schisandraceae (SC), and*Withania somnifera* (L.) Dunal, Solanaceae (WS)

Comply with the key criterium, that is to exhibit multitarget effects on the neuroendocrine-immune system by triggering adaptive stress responses. These include stimulation of cellular and organismal defense systems, activation of intracellular and extracellular adaptive signaling pathways, and expression of stress-activated proteins to change protection or repair capacity and increase non-specific resistance and adaptation to stress [11,12].

Anti-inflammatory, antiviral, antioxidant and other related activity of plants referred to as adaptogenic have been demonstrated in numerous pre-clinical studies. Table 1, Table 2, Table 3, Table 4 and Table 5 and Figure 1 show multiple molecular targets identified for adaptogenic plant extracts exhibiting:Specific antiviral action-preventing viruses binding to host cells, and on non-structural (Nsps) and structural proteins involved in viral life cycle in infected host cells and replication of the virus;Non-specific antiviral action by the effects on:
○Innate immunity including activation of defensins, the complement system, upregulation of expression of pathogen’s pattern recognition receptors, specifically TLR, and interferons;○Downregulation of expression of pro-inflammatory cytokines IL-1, IL2, IL-6, IL-8, and TNF, activation of natural killer cells, mucous sentinel and phagocyting cells (mast cells, dendritic cells, macrophages, neutrophils, eosinophils, and basophils) and the melatonin signaling pathways;○Adaptive immunity including T cells and MHC proteins, B cells and antibodies.Anti-inflammatory activity by inhibition of:
○Release of arachidonic acid from membrane phospholipids following conversion into COX-2 and lipoxygenase-mediated pro-inflammatory metabolites such as prostaglandins, thromboxane B2, leukotrienes, as well as platelet-activating factor;○Inducible NO synthase;○NF-κB-mediated pro-inflammatory signaling pathways.Detoxifying and cytoprotectant activity in oxidative stress-induced injuries of compromised cells and tissues:
○Activation of the NRF2-mediated oxidative stress response signaling pathway regulated production of chaperons and stress response proteins, activity of phase I and II metabolizing enzymes, phase III detoxifying proteins, proteasomal degradation proteins, antioxidant proteins (superoxide dismutase (SOD), glutathione S-transferase (GST), NAD(P)H quinone oxidoreductase 1 (NQO1) and heme oxygenase 1 (HO1);○Activation of expression and release of molecular chaperons Hsp70, which mediate cytoprotectant and repair processes;Activation of the melatonin signaling pathways.

Key elements of innate immunity stimulation are activation of first-line defense response IF-γ and TLR followed by inhibition of NF-κB and inflammation mediated by proinflammatory cytokines. Adaptogens activate adaptive signaling pathways by upregulating gene expression-encoding phosphatidylinositol 3-kinase (PI3K), protein kinase C (PKC), and mitogen-activated protein kinases (MAPKs) [11], which are upstream of transcription factors (Nrf2, HNF1, CCAAT, C/EBPβ, and PXR), FXR and peroxisome proliferator-activated receptors that promote the induction of phase II enzymes and phase III transporters involved in metabolic detoxification process, clearance of breakdown products [16] and overall defense response to pathogens. Therapeutically important features of adaptogenic activity are beneficial effects on detoxification, and repair processes, leading to recovery and increased survival in virus-induced oxidative stress, key to which are activation of the antioxidant NRf2-signalling pathway, the production of detoxification enzymes, molecular chaperons Hsp70 and the melatonin signaling pathway for regulation of homeostasis (Table 3 and Table 4).

The search for anti-SARS-CoV-2 therapeutics focuses on both structural and functional viral proteins. Sixteen non-structural proteins (Nsps 1–16) are involved in RNA transcription, translation, protein synthesis, processing and modification, virus replication and infection of the host [17]. They are considered virus-specific molecular targets for pharmacotherapeutic intervention [17,18,19,20,21,22,23] for a number of reasons:N-terminal gene 1 protein (Nsp1) suppresses host innate immune response, inhibiting type-I interferon production and induces host mRNA degradation [24];Nsp3 (papain-like protease, PLpro) is essential for virus replication and to antagonize the host’s innate immunity;Nsp5 (3-chimotrypsin-like protease, 3CLpro) mediates viral replication, transcription and the maturation of Nsps, which is essential in the life cycle of the virus;Nsp12 (PNA-dependent PNA polymerase enzyme, RdRp) is a conserved vital enzyme of the coronavirus replication/transcription complex;Nsp13 (helicase enzyme) is a multifunctional protein necessary for the replication of coronavirus.

Nine structural and accessory proteins, including spike (S) and envelope (E) glycoproteins, membrane (M) and nucleocapsid (N) proteins, are probably most important in the search for inhibitors of their expression or functions [19]. The primary function of structural S proteins is to bind the S1 subunit with the host cell surface receptor, angiotensin-converting enzyme 2 (ACE2), and the S2 subunit with serine protease TMPRSS2, which mediates virus–cell and cell–cell membrane fusion. ACE2 has been identified as a functional receptor playing a crucial role in SARS coronavirus-induced lung injury [25,26]. ACE2 is expressed in all tissues—particularly in pulmonary and heart tissues, where it is significantly increased in hypertensive patients continuously using ACE inhibitors and angiotensin 1 receptor (AT1R) blockers. This explains the higher death rate in elderly individuals with comorbidities such as hypertension, diabetes, and heart disease [27,28,29,30].

## 2. Results

### 2.1. Pre-Clinical Investigations

We have organized results of pre-clinical investigations into five groups: direct viricidal effects, specific antiviral actions, non-specific antiviral actions, anti-inflammatory effects and repair of oxidative stress-induced injuries in compromised cells and tissues, and other effects of potential relevance in the progression of viral infections. Outcomes are presented in Table 1, Table 2, Table 3, Table 4 and Table 5. They include results for aforementioned adaptogenic plant extracts and some of their active constituents: tetracyclic and pentacyclic triterpene glycosides (ginsenosides, withanolides, and glycyrrhizin), diterpene lactones (andrographolides), phenethyl- and phenylpropanoid glycosides (salidroside, rosavin, and eleutheroside B), and lignans (eleutheroside E, schisandrins, anwulignan, and ellagic acids).

### 2.2. Clinical Investigations

#### 2.2.1. *Andrographis paniculata*

Results of 33 RCTs (7175 patients) with AP (as a monotherapy and as fixed combinations with other herbs) clinical studies were systematically reviewed. The meta-analysis shows that AP improved cough (n = 596, standardized mean difference SMD: −0.39, 95% confidence interval CI [−0.67, −0.10]) and sore throat. It has a statistically significant effect in improving overall symptoms of acute respiratory tract infections (ARTIs) when compared to placebo, usual care, and other herbal therapies. Evidence also suggested that AP (alone or plus usual care) shortened the duration of cough, sore throat and sick leave/time to resolution when compared versus usual care. No major adverse events (AEs) were reported, and minor AEs were mainly gastrointestinal [242].

Efficacy and safety of andrographolide-containing preparations was studied in patients with common cold in Scandinavia, South America, and India by Hancke et al. [243], Caceres et al. [244], Melchior et al. [245], and Saxena et al. [246]. These four randomized double-blind placebo-controlled trials cover in total 539 patients suffering from symptoms of common cold. Hancke et al. [243] found the efficacy and safety of AP tablets (1200 mg/day) to be superior to placebo. The intensity of symptoms and signs of rhinitis, sinus pain and headache were significantly lowered compared to placebo. No adverse events were reported. Melchior et al. [245] performed a randomized, double-blind, placebo-controlled, monocenter, parallel-group trial with AP (1020 mg/kg) in 50 patients suffering from common cold over 5 days. The sick leave days were significantly reduced after the second visit in the verum group compared to placebo, the number of patients feeling recovered was increased and the number of patients experienced with easier disease was detected also to be better than placebo. Caceres et al. [244] also tested the treatment of common cold with AP (1200 mg.dat) in a randomized, double-blind, placebo-controlled, monocenter, parallel-group trial with 158 participants over 5 days. The intensity of nearly all symptoms decreased significantly in the verum group. The active treatment was clearly superior to placebo, reducing the prevalence and intensity of symptoms without observed or reported adverse effects and thus revealed a positive benefit/risk ratio. Saxena et al. [246] tested an AP extract (200 mg/day, 60 mg of andrographolide for 5 days) in a randomized, double-blind placebo-controlled clinical study involving 223 patients with uncomplicated upper respiratory tract infections. Only in the verum group all symptoms improved significantly (*p* ≤ 0.05) except earache. The overall efficacy of KalmCold™ over placebo was 2.1-fold higher (*p* ≤ 0.05) than placebo.

#### 2.2.2. *Eleutherococcus senticosus*

Several epidemiological studies carried out in the Soviet Union during the 1970s demonstrate that ES extract, given prophylactically, can reduce human mortality rates during the influenza epidemics as well as typical complications of an influenza infection, such as pneumonia, bronchitis, and otitis [221,247,248,249].

In 1986, Shadrin et al. [250] reported the results of prophylactic treatment of 1376 patients with acute respiratory illnesses during the influenza virus epidemic. Typical complications of an influenza infection, such as pneumonia, bronchitis, genyantritis and otitis, were determined in this two-parallel-group, placebo-controlled, double-blind study with a 3-month long follow-up period. A significantly lower frequency of complications caused by infections was observed in the ES group compared to the placebo group (*p* < 0.05), indicating milder infection progression. The overall morbidity rate was also consistently lower in the ES group than in the placebo group, but the differences were not statistically significant. Two consecutive open-label clinical studies of ES extract were carried out in 764 children with respiratory viral infections. The morbidity rate decreased 3.6-fold in those 396 children treated with ES liquid extract for a month. After 2 years, a 2–3-fold lower morbidity was recorded in those receiving ES liquid extract for a month compared to the control group of 252 children [251]. In a similar study with children at pre-school age, prophylactic administration of ES extract decreased the morbidity rate by 30–40% [252].

#### 2.2.3. *Glycyrrhiza* spp.

Clinical trials conducted with GS have focused functional dyspepsia, aphthous stomatitis, gastric and duodenal ulcers, postoperative sore throat, hyperlipidemia and antiatherogenic effects [207]. Since the publication of the EMA’s assessment report, numerous further clinical trials have been conducted. A recent review by Kwon et al. [253] summarizes study results related to liver, gastrointestinal, oral, skin and metabolic disorders which confirm licorice’s anti-inflammatory, antioxidant, and immunomodulatory properties. However, the authors caution against chronic use, especially in patients with cardiovascular comorbidities due to the mineralocorticoid-like effect of glycyrretinic acid and GS-induced pseudoaldosteronism.

#### 2.2.4. *Panax* spp.

PS has been extensively studied in clinical investigations of multiple adaptogenic indications [208]. Scaglione et al. [254] conducted a clinical trial of efficacy and safety of a PS extract for potentiating vaccination against the common cold and/or influenza syndrome in 227 volunteers and reported a significantly lower frequency of influenza or common cold the treatment group. The same group [255] reported significantly increased bacterial clearance in patients with chronic bronchitis who received PS extract concomitantly with antibiotic treatment. Lee et al. [256] conducted a clinical trial investigating the preventive activity of PS against acute respiratory illness (ARI) caused by viral infection in 100 volunteers and found ginseng to protect against contracting ARI, as well as decrease the duration and scores of ARI symptoms. Iqbal and Rhee [112] reviewed the evidence for antimicrobial activity of PS, specifically against pathogens causing respiratory infections from animal and in vitro models, as well as 15 clinical trials. Summarily, included investigations have shown PS to exert immunomodulatory activity, which reduces the level of proinflammatory cytokines and oxidative stress, which, in turn, reduce severity, duration, and frequency of symptoms and show potential for preventing development of respiratory infections.

#### 2.2.5. *Rhodiola rosea*

Multiple clinicals trials on the effect of RR as a mono-product and in combinations on physical performance and stress-related fatigue have been conducted [209], affirming the traditional use as an adaptogen. Chuang et al. [257] studied the effect of RR as an adjunct treatment in patients with Chronic Obstructive Pulmonary Disease (COPD) and found it to significantly improve tidal breathing and ventilation efficiency. Zhang et al. [258] evaluated the effects of RR on the preventive treatment of acute lung injury (ALI) caused by post-traumatic/inflammatory and thoracic-cardiovascular operations. They observed a significant decrease in Acute Respiratory Distress Syndrome complications and concluded that early use of RR may protect against risk factors of ALI/ARDS. This recommendation was later confirmed by Lu et al. [259] in a similar trial. Ahmed et al. [260] studied the antiviral properties of RR in marathon runners. RR induced antiviral activity early and delayed exercise-dependent increase in virus replication. RR’s role in the treatment of ischemic heart disease was investigated by Yu et al. [261] in a meta-analysis of 13 clinical trials and found an overall positive effect on both improvement of symptoms and ECG.

#### 2.2.6. *Schisandra chinensis*

Pre-clinical findings have been corroborated in numerous clinical investigations, specifically SC’s effect in viral respiratory tract infections [212,250,262,263,264], by targeting viral RNA synthesis and replication and stimulating innate and adaptive immunity, among others.

#### 2.2.7. *Withania somnifera*

Tandon and Yadav [265] reviewed 30 human clinical trials, establishing reasonable safety and efficacy in subclinical hypothyroidism; chronic stress, insomnia and anxiety; cognitive improvement; fertility; and as a chemotherapy adjuvant, among others. Adaptogenic effects were studied in three clinical trials, one of which [266] reported significantly increased oxygen consumption, maximum velocity, and average absolute and relative power under exercise conditions with WS supplementation, an outcome that may be relevant in convalescence from respiratory disease.

#### 2.2.8. Combination Products

The results of five randomized, double-blind placebo-controlled studies with a fixed combination of AP and ES (Kan Jang, KJ) conducted between 1997 and 2004 in Scandinavia, South America, Russia and Armenia suggest that it relieves symptoms of uncomplicated respiratory tract infections caused by common cold [267,268,269,270,271] without causing any safety concerns. Caceres et al. [267] investigated the prevention of common cold in a randomized, double-blind, placebo-controlled, monocenter, parallel-group trial with 107 participants over 3 months. KJ showed a significantly reduced incidence rate of cold compared to placebo after three months. Melchior et al. [270] investigated KJ in two randomized, double-blind, placebo-controlled, monocenter, parallel-group, pilot and phase III clinical trials with correspondingly 46 and 179 participants for a maximum of 8 days (pilot study) and followed by a phase III study for 3 days in the treatment of uncomplicated upper respiratory tract infections. In the pilot study, the active therapy by Kan Jang was superior to placebo in the reduction in the total scores for all symptoms after 5 days. In the phase III study, the symptom score was more significantly improved in the treatment group compared to placebo. Gabrielian et al. [268] investigated KJ in a double-blind, placebo-controlled, multicenter, parallel-group trial in 185 participants with acute upper respiratory tract infections including sinusitis over 5 days. KJ was found to be a valuable therapeutic option and to have a positive benefit/risk ratio for the treatment of acute upper respiratory tract infections and for relief of inflammatory symptoms of sinusitis. Kulichenko et al. [269] investigated KJ in two randomized, comparator-controlled, open multicenter, parallel-group trials with 540 participants over 3 to 5 days in a pilot study, followed by a phase III study for 5 days in the treatment of uncomplicated upper respiratory infections. KJ was found to be superior in alleviation of symptoms such as headache, myalgia and conjunctivitis. Spasov et al. [271] investigated KJ in a randomized controlled three parallel-group study in 130 children with uncomplicated common cold over a period of 10 days. The amount of nasal secretion g/day and nasal congestion g/day and nasal congestion decreased significantly, and recovery time was significantly accelerated by KJ compared to Immunal and standard therapy. Kan Jang was well tolerated, and no side effects were observed in this group.

The postmarketing pharmacovigilance assessment of KJ shows a favorable benefit/risk ratio. Only 37 adverse event reports (mainly to allergic reactions) to the Swedish and Danish competent authorities were recorded in 23 years with over 20 million doses of KJ sold. This equates to one adverse event in approximately 100,000 patients, assuming an average drug uptake for 5 consecutive days [272].

#### 2.2.9. Activation of the Melatonin Signaling Pathway

Another promising tool to non-specifically curb SARS-induced progression of inflammation, particularly in elderly subjects, with adaptogens is to utilize their capacity to activate the melatonin signaling pathway.

In a recent study of the molecular mechanisms of action of adaptogens, it was found that they activate the melatonin signaling pathway by acting through two G protein-coupled membrane receptors MT1 and MT2 and upregulation of the ligand-specific nuclear receptor gene RORA [11] which encodes retinoic acid receptor (RAR)-related orphan nuclear receptor alpha (RORα)—a multifunctional transcription activating factor involved in many physiological processes, including regulation of immunity and metabolism, as well as playing an important role in several pathologies, including inflammation, autoimmune diseases, asthma, osteoporosis, cancer, and metabolic syndrome [202,273,274,275].

Furthermore, the molecular mechanism of actions of melatonin [276,277] and adaptogens are alike in terms of their effects on expression of many genes including UCN, GNRH1, TLR9, GP1BA, PLXNA4, CHRM4, GPR19, VIPR2, RORA, STAT5A, ZFPM2, ZNF396, FLT1, MAPK10, MERTK, PRKCH, and TTN, suggesting that melatonin is an adaptation hormone [278,279], playing an important role in regulation of homeostasis [11]. This conclusion is in line with the common opinion about its physiological role and functions with pleiotropic actions in human, animals, and plants, which include controlling senescence and aging, regulating circadian rhythms, defense response to pathogens and bolstering the immune system [202,277,280,281,282,283].

The concentration of melatonin in human serum significantly increases at nighttime from 15–20 to 30–180 pg/mL. However, with age, the level of night melatonin does not increase higher than 30 pg/mL [284]. The decreased ability to produce melatonin with aging is probably associated with low-grade chronic inflammation and aging-related diseases. Melatonin has been shown to exert anti-inflammatory, antioxidant, and other beneficial actions in aging [202].

Melatonin has been found not only in humans, but also in bacteria, mammals, birds, amphibians, reptiles, fish, and plants. The richest plant sources of melatonin are *Coffea* spp. with 5800–6500 ng/g [285], *Tanacetum parthenium* (L.) Sch. Bip. with 2450 ng/g, *Viola philipica* Cav. with 2368 ng/g, *Uncaria rhynchophylla* (Miq.) Jacks. with 2460 ng/g, *Hypericum perforatum* L. with 4390 ng/g, and *Morus alba* L. with 1510 ng/g, to name just a few [283,286]. Some adaptogens also contain melatonin in amounts of 100–500 mg/g, e.g., AP with 511 ng/g, PS with 169 ng/g, SC with 86 ng/g, and GS with 112 ng/g [283,286], and therefore effects of adaptogens on the melatonin signaling pathways may in part be due to its presence. However, other adaptogenic plants, such as RR, ES, and WS, do not contain melatonin, but nonetheless significantly activate the melatonin signaling pathway and upregulate RORA expression [11].

While the content of melatonin in the plants studied ranged from 0.01 to 6500 ng/g dry weight [283,286,287,288,289], assuming a therapeutic daily dose in the range from 1 to 10 g of dry herbal substance, the consumed melatonin (<0.065 mg) would be significantly lower than its therapeutic daily dose of 3–10 mg. However, these contents are in the range of daily amount (30 μg) of melatonin synthesized in adult humans. Since oral bioavailability of melatonin is approximately 3% in 27-year-old heathy subjects [290], the consumption of 10 g of dry herbal substance containing 1000 ng/g of melatonin can increase its concentration in blood to approximately 12 pg/mL, which is comparable to physiological concentrations of endogenous melatonin in the blood, particularly in elderly people 60–70 years of age, from 15 pg/mL at daytime to 30 pg/mL at nighttime [284].

Recently, melatonin was prioritized as a potential SARS-CoV-2 repurposable drug using network pharmacology-based methodologies that quantify the interplay between the virus–host interactome and drug targets in the virus–host protein interactions network [23,291]. This conclusion aligns with results of other studies summarized in several reviews [292,293,294]. For instance, melatonin was found to exhibit therapeutic potential in influenza A H1N1 virus infection, to elicit anti-inflammatory and immune modulatory effects—the induction of IL-10 by melatonin occurs via the upregulation of IL-27 in DC—and to exhibit a synergistic effect with an antiviral drug [295]. In another study, intranasally inoculating mice with RSV resulted in oxidative stress changes by increasing NO, MDA and -OH levels, and decreasing GSH and SOD activities. Administration of melatonin significantly reversed all these effects. Furthermore, melatonin inhibited the production of proinflammatory cytokines such as TNFα in serum of RSV-infected mice. These results suggest that melatonin ameliorates RSV-induced lung inflammatory injury in mice via inhibition of oxidative stress and proinflammatory cytokine production and may indeed be considered a novel therapeutic agent in virus-induced pulmonary infection [296]. Melatonin was also found to exert direct viricidal effects against respiratory syncytial virus [296,297], Semliki Forest virus [298] and Venezuelan equine encephalomyelitis virus [299,300,301].

These effects may be partially associated with melatonin-induced upregulation of RORA encoding RORα in the liver, thymus, brain, skeletal muscle, skin, lung, adipose tissue, and kidney. RORα has an anti-inflammatory function in human macrophages. In RORA-deleted cells, a dramatic increase in basal expression of a subset of NF-κB-regulated genes, including TNF, IL-1β and IL-6, at both transcriptional and translational levels was observed [302]. The expression of RORα1 inhibits TNFα-induced IL-6, IL-8 and COX-2 expression in primary smooth-muscle cells [303] and plays an essential role in the development of type 2 innate lymphoid cells (ILC2) and in cytokine production in the ILC3 and Th17 cells [274,304,305,306]. While RORα is a known inhibitor of NF-κB proinflammatory signaling, it can also be utilized by highly pathogenic influenza H5N1 virus, which can inhibit inflammatory response in monocytes via activation of RORα and therefore prevent an effective inflammatory defense response of monocytes [307].

## 3. Discussion

From the data presented here, it can be concluded that adaptogens can play a potentially important role at all stages of viral infection. Adopting a recently proposed phased immunophysiological approach to viral infection [308], adaptogens exerting multitarget effects on the neuroendocrine-immune system by triggering adaptive stress responses have a place in prevention, infection, escalating inflammation and recovery (Figure 2). They provide baseline support through their immunomodulatory, immunostimulatory, and antioxidant effect through all four phases, combat infection through their specific and non-specific antiviral properties, alleviate escalating inflammation through their anti-inflammatory effects, as well as their capacity to repair oxidative stress-induced injuries in compromised cells and tissues, and address secondary disease states and comorbidities through various, infection-related activities.

Another possible benefit of adaptogens in COVID-19 is their effect during convalescence of patients. This is based on the results of an RCT with Chisan/ADAPT-232, a fixed combination of ES with RR and SC in pneumonia [309]. Adjuvant therapy with ADAPT-232 had a positive effect on the recovery of patients by decreasing the duration of the acute phase of the illness, by increasing mental performance of patients in the rehabilitation period, and by improving their quality of life.

## 4. Materials and Methods

Where available, assessment reports published by the European Medicines Agency (EMA)’s Herbal Medicinal Products Committee (HMPC) were consulted to provide the summaries below. To cover the time elapsed since publication of these reports and in all other cases, database searches in PubMed, Scopus and Google Scholar were performed. Keywords included scientific and common plant names, in combination with “antiviral”, “adaptogen *”, “respiratory”, “human”, and “clinical trial”. Relevant primary literature referenced in reviews was retrieved manually. Independent searches were performed regarding relevant information on SARS-CoV-2 and COVID-19, as well as the evidence for therapeutic relevance of phytomedicines in COVID-19.

## Figures and Tables

**Figure 1 pharmaceuticals-13-00236-f001:**
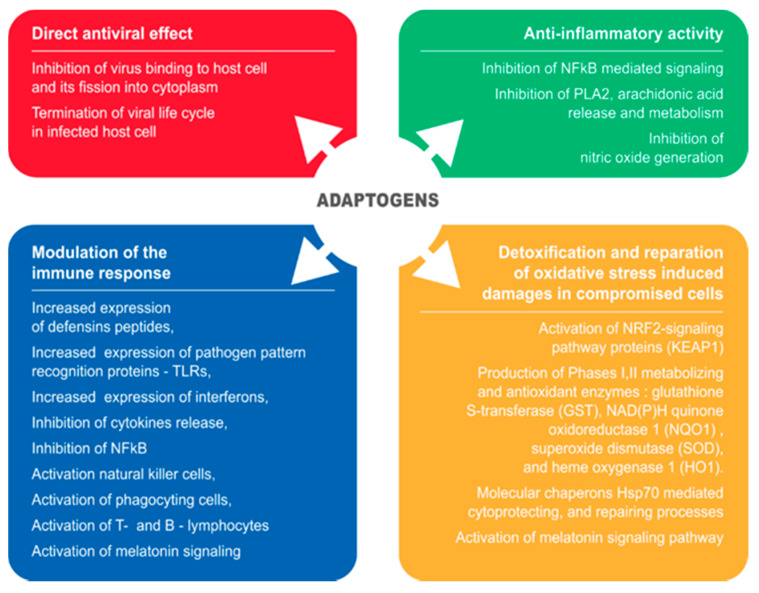
Schematic diagram of reported effects of adaptogenic plants elucidated in animal and cell culture models: (i) modulatory effects on immune response (blue block), (ii) anti-inflammatory activity (green bock), (iii) detoxification and repair of oxidative stress-induced damage in compromised cells (brown block), and (iv) direct antiviral effect via infraction with viral docking or replication (red block).

**Figure 2 pharmaceuticals-13-00236-f002:**
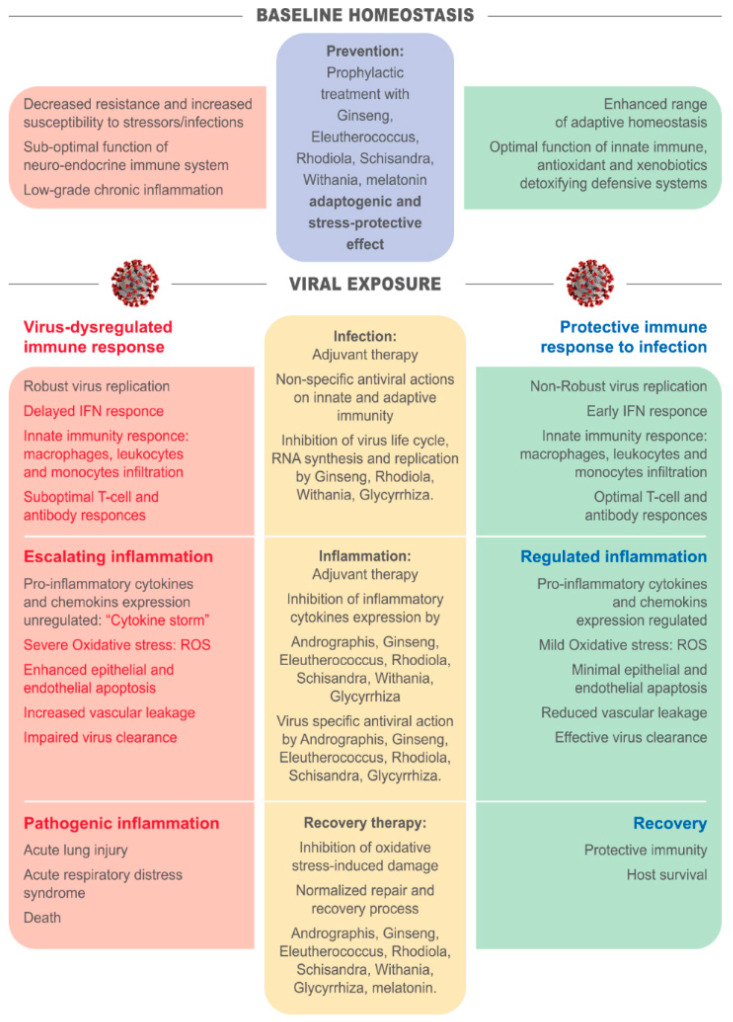
Schematic diagram of various phases of immune and inflammatory responses to SARS-CoV-2 infection and stages of COVID-19 progression with and without considering potential effects of adaptogenic plants on prevention, infection, inflammation, and recovery phases of viral infection.

**Table 1 pharmaceuticals-13-00236-t001:** Direct viricidal effects.

Virus	AP Andrographolides	ES Eleutherosides	GS Glycyrrhizin and Glycyrrhizic Acid	PS Ginsenosides	RR Salidroside, Rosavin, Ellagic and Gallic Acids	SC Schisandrins and Anwulignan	WS Withanolides
**SARS-related coronavirus**			[31,32,33]				
**Ebola virus (EBOV) and Marburg virus (MARV)**					[34]		
**Human rhinovirus (HRV)**		[35]					
**Respiratory syncytial virus (RSV)**		[35]	[36]	[37]			
**H1N1 influenza A virus**	[38,39,40]	[35,41,42,43,44]	[40,45]	[46,47,48,49,50,51,52,53]	[54]		
**H2N2 influenza virus**			[55]				
**H3N2 influenza virus**				[50,53,56]			
**H5N1 avian influenza virus**	[57]		[58]	[50,56]	[34]		
**H7N9 influenza**				[50]			
**H9N2 avian influenza virus**					[54]		
**Chikungunya virus**	[59,60,61]						
**Dengue virus**	[59,60]				[62]		
**Coxsackievirus B3**					[63]		[64]

**Table 2 pharmaceuticals-13-00236-t002:** Specific antiviral actions/effects on SARS virus docking and replication.

Target/Mediator	AP Andrographolides	ES Eleutherosides	GS Glycyrrhizin and Glycyrrhizic Acid	PS Ginsenosides	RR Salidroside, Rosavin, Ellagic and Gallic Acids	SC Schisandrins and Anwulignan	WS Withanolides
**Effects on Viral Life Cycle in Infected Host Cells—Targets Preventing the Virus RNA Synthesis and Replication**							
**Nsp5: 3-chymotrypsin-like protease (3Clpro)–main protease of SARS-CoV-2 (Mpro)**	[22,65]	[19,66]		[19,66]	[19,66]	[19,66]	
**Nsp3: Papain-like protease (Plpro)**	[22]	[19,66]		[19,66]	[19,66]	[19,66]	
**Nsp12: RNA-dependent RNA polymerase (RdRp)**	[22]						
**Nsp1: The most N-terminal gene 1 protein**	[22]						
**Targets Inhibiting Virus Structural Proteins**							
**S1: Spike glycoprotein binding SARS-CoV-2 to human angiotensin-converting enzyme 2 (ACE2) of host cells**			[67]				
**S2: Spike glycoprotein receptor to type-II transmembrane serine protease enzymes (TMPRSS2) of host cells**	[22]		[67]				
**Blockage of binding viral (Ebola/Marburg) surface glycoproteins to host cells**					[34]		

**Table 3 pharmaceuticals-13-00236-t003:** Non-specific antiviral actions/effects on innate and adaptive immunity.

Target/Mediator	AP Andrographolides	ES Eleutherosides	GS Glycyrrhizin and Glycyrrhizic Acid	PS Ginsenosides	RR Salidroside, Rosavin, Ellagic and Gallic Acids	SC Schisandrins and Anwulignan	WS Withanolides	Melatonin
**Innate Immunity**								
**Defensins: Human β-defensin-2**	[68,69]							
**Pathogen’s pattern recognition via TLR**	[70,71]	[11,72]	[67,73,74,75,76,77]	[78,79,80,81,82,83]	[11,84,85]	[11,86,87,88]	[11,89]	[90,91,92,93,94]
**Interferons**	[95]	[96,97,98,99,100,101,102]	[33,55]	[37,46,48,49,56]	[62,63]	[86]	[103,104,105,106]	[90]
**Natural killer cells**		[97,98]	[58]	[46]				
**Interleukins: IL-6, IL-1** **β** **, IL-10, TNF, etc.**	[95,107,108]	[108,109]	[74,77,110]	[37,48,49,52,111,112]	[62,63,84]	[86,110,113]	[103,104,105,106,114]	[90,92,94,115]
**Melatonin signaling pathways**		[11]			[11]	[11]	[11]	[11]
**Adaptive Immunity**								
**T cells and MHC proteins**		[97,98,99,116]	[33,55,58]			[86]	[103,104,105,106,117]	
**B cells and antibodies**	[95]	[72]	[55]				[104,105,106]	

**Table 4 pharmaceuticals-13-00236-t004:** Anti-inflammatory effects and repair of oxidative stress-induced injuries in compromised cells and tissues.

Target/Mediator	AP Andrographolides	ES Eleutherosides	GS Glycyrrhizin and Glycyrrhizic Acid	PS Ginsenosides	RR Salidroside, Rosavin, Ellagic and Gallic Acids	SC Schisandrins and Anwulignan	WS Withanolides	Melatonin
**Arachidonic acid release, inhibition of phospholipase 2**	[118]		[119,120,121]	[122,123,124,125]	[126]	[127]	[128,129,130]	
**COX-2-mediated signaling**	[71,131]	[132]	[74,119,121]	[111,123,125]	[132]		[132]	[132]
**Lipoxygenase-mediated signaling of arachidonic acid pro- and anti-inflammatory metabolites leukotrienes, lipoxins, resolvins, etc.**		[132]	[119,121]		[132]	[127]	[130,132]	[132]
**Platelet-activating factor (PAF)**	[133,134]			[135,136]		[137,138]	[89]	
**Nitric oxide-mediated inflammation:** **Inducible NO synthase** **oxide catabolites (NOCs)**	[108,139]	[108]	[32,74,108]	[52]	[140]	[88,108,140]		[90,115]
**NF-κB-mediated inflammation** **NF-κB signaling, translocation and expression**	[70,71,107,141]	[72,142,143,144,145,146,147,148,149,150]	[74,77]	[151,152,153,154,155]	[156,157,158,159,160,161,162,163]	[86,87,113,164,165,166,167,168]	[89,114,169,170,171,172]	[90,173]
**Nrf2-mediated oxidative stress response signaling pathway proteins:** **Phosphatidylinositol 3-kinase (PI3K), protein kinase B (Akt), KEAP1, etc.** **Nrf2-ARE (antioxidant response element) expression**	[174,175,176,177,178,179,180,181,182]	[183]	[119]	[111,151,184,185,186,187]	[159,160,162,163,188,189,190]	[86,113,165,191,192,193]	[194,195,196]	[197]
**Antioxidant proteins (SOD, GST, NQO1 and HO1), lipid peroxidation**	[176,178,182]	[183]		[111,151,185,187,198]	[158,160,162,163,188,190]	[86,113,165,191,192,193]	[103,194]	[93,115,197]
**Molecular chaperon-mediated cytoprotectant and repair processes Heat shock proteins Hsp72**		[199,200]			[140,199,200,201]	[140,199,200,201]		
**Melatonin signaling** **Retinoic acid receptor (RAR)-related orphan nuclear receptor alpha (RORα)**		[11]			[11]	[11]	[11]	[11,202,203]

**Table 5 pharmaceuticals-13-00236-t005:** Other effects of potential relevance in the progression of viral infections.

Activity	AP	ES	GS	PS	RR	SC	WS
**adaptogenic**	[204,205]	[206]	[207]	[208]	[209]	[206,210,211,212,213]	[214,215,216]
**antidiabetic**	[217,218,219,220]	[221]				[212]	[214,215,216]
**antioxidant**	[222,223,224]	[221]	[207]	[208]	[209]	[210,211,213]	[214,215,216]
**immunomodulatory**	[225]	[221]	[207]	[208]		[210,211,213]	[214,215,216]
**metabolism**			[207]	[208]	[209]	[212]	
**gastroprotective**	[226,227]		[207]			[212]	
**hepatoprotective**	[228,229,230,231]	[221]	[207]	[208]		[210,211,213]	[214,215,216]
**cardioprotective**	[232,233]	[221]		[208]	[209]	[210,211,213]	[214,215,216]
**antiproliferative**	[234,235,236]	[221]	[207]	[208]	[209]	[210,211,213]	
**neuroprotective**	[237]	[221]		[208]	[209]	[210,211,213]	[214,215,216]
**anti-stress/anti-fatigue**	[238]	[221]		[208]	[209]	[210,211,213]	[214,215,216]
**antidepressant**	[239]	[221]	[207]	[208]	[240,241]	[212]	[214,215,216]

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
