# Peer review of "The Role of Adaptogens in Prophylaxis and Treatment of Viral Respiratory Infections"

_pharmaceuticals, 2020, doi:10.3390/ph13090236_

Round 1

Reviewer 1 Report

-In line 356 where it says "TNF-a" should it be "TNF-∝"

-In some references (for example in numbers 6, 21, 244) the DOI is missing.

-Although the title and abstract and the rest of the article talks about viral infections, the introduction of the article is heavily centred in SARS-CoV-2, and in a way, it could be a little bit misleading regarding the data reviewed in the article.

-Maybe it should be interesting to address the issue of the fact that, when using and/or evaluating plant extracts, it is difficult to attribute the activity to a molecule (in extracts there are many molecules that could create synergies, and also the presence and the amount of the existing molecules depend on several factors during the harvest, such as temperature, season, date,... or even potential artefacts due to the solvents used) hence the difficulty of using them as drugs.

-Also, it could be interesting to name the type of compounds present in the plants selected in this review.

Reviewer 2 Report

The article is a clear material dealing with the role of adaptogens in the prevention of viral infection. Currently, this issue is very topical and the article brings very encouraging insights and information. A large amount of literature is cited in the article. It contains two clear diagrams about the effects of adaptogens. From my point of view, no text correction is necessary.
